# An Oncolytic Adenovirus Encoding SA-4-1BBL Adjuvant Fused to HPV-16 E7 Antigen Produces a Specific Antitumor Effect in a Cancer Mouse Model

**DOI:** 10.3390/vaccines9020149

**Published:** 2021-02-12

**Authors:** Alejandra G. Martinez-Perez, Jose J. Perez-Trujillo, Rodolfo Garza-Morales, Norma E. Ramirez-Avila, Maria J. Loera-Arias, Jorge G. Gomez-Gutierrez, Odila Saucedo-Cardenas, Aracely Garcia-Garcia, Humberto Rodriguez-Rocha, Roberto Montes-de-Oca-Luna

**Affiliations:** 1Department of Histology, School of Medicine, Autonomous University of Nuevo León, Monterrey, NL 64460, Mexico; alejandra.martinezpr@uanl.edu.mx (A.G.M.-P.); jperez.me0052@uanl.edu.mx (J.J.P.-T.); nramireza@avantsante.com (N.E.R.-A.); mdjesus.loeraars@uanl.edu.mx (M.J.L.-A.); odila.saucedocr@uanl.edu.mx (O.S.-C.); aracely.garciagr@uanl.edu.mx (A.G.-G.); humberto.rodriguezrc@uanl.edu.mx (H.R.-R.); 2Department of Surgery, School of Medicine, University of Louisville, Louisville, KY 40202, USA; rodolfo.garzamorales@utrgv.edu (R.G.-M.); jgguti01@louisville.edu (J.G.G.-G.); 3Department of Molecular Genetics, Northeast Biomedical Research Center, Mexican Institute of Social Security (IMSS), Monterrey, NL 64000, Mexico

**Keywords:** 4-1BBL, cancer vaccines, oncolytic virus, HPV-16 antigen

## Abstract

Human papillomaviruses (HPVs) are responsible for about 25% of cancer cases worldwide. HPV-16 E7 antigen is a tumor-associated antigen (TAA) commonly expressed in HPV-induced tumors; however, it has low immunogenicity. The interaction of 4-1BBL with its receptor induces pleiotropic effects on innate, adaptive, and regulatory immunity and, if fused to TAAs in DNA vaccines, can improve the antitumor response; however, there is low transfection and antitumor efficiency. Oncolytic virotherapy is promising for antitumor gene therapy as it can be selectively replicated in tumor cells, inducing cell lysis, and furthermore, tumor cell debris can be taken in by immune cells to potentiate antitumor responses. In this study, we expressed the immunomodulatory molecule SA-4-1BBL fused to E7 on an oncolytic adenovirus (OAd) system. In vitro infection of TC-1 tumor cells and NIH-3T3 non-tumor cells with SA/E7/4-1BBL OAd demonstrated that only tumor cells are selectively destroyed. Moreover, protein expression is targeted to the endoplasmic reticulum in both cell lines when a signal peptide (SP) is added. Finally, in an HPV-induced cancer murine model, the therapeutic oncolytic activity of OAd can be detected, and this can be improved when fused to E7 and SP.

## 1. Introduction

Cancer ranks among the leading causes of mortality with approximately 8 million deaths worldwide registered in 2015 [1], predominately in low and medium socioeconomic countries. Around 25% of all cancer cases are caused by oncogenic viral infections such as human papillomavirus (HPV) 16 and 18 serotypes [2].

Current treatments are surgery, radiation therapy, and chemotherapy, resulting in efficient tumor clearance. Unfortunately, these strategies lead to adverse effects that affect the patient’s quality of life, a high tumor recurrence rate, and development of resistance to chemotherapy. Thus, new therapeutic approaches have been explored. Gene therapy has been used as a vaccine technology that employs naked DNA or vectors (viral and non-viral) to express modified recombinant antigens and immunostimulant molecules to elicit a specific immune response capable of eliminating tumor cells [3].

HPV-16 E7 antigen is an ideal tumor-associated antigen (TAA) that has been extensively studied in preclinical vaccine research models [4]. Apart from the MHC–TCR interaction during antigen presentation, it is crucial that the binding interaction of immunostimulatory co-receptors is also present; otherwise, poor or no activation of the cell immune response is obtained. Among the immunostimulant molecules, 4-1BBL is a ligand that is expressed in several antigen-presenting cells (APCs), such as B lymphocytes, macrophages, and dendritic cells, as well as in activated T cells [5]. 4-1BBL interacts with its high-affinity 4-1BB receptor, which is overexpressed on T cells that have been activated by APCs or co-stimulatory agonists, and it can also be found on CD11c+ dendritic cells. Among the T cell subsets are the CD4+, CD8+, NK, NKT, and CD4+ CD25+ regulatory cells (Treg) [6], and such an interaction induces T-cell clonal expansion, survival, and the establishment of long-term immune memory [7,8].

Several reports have demonstrated that specific antitumor immune responses can be elicited using naked DNA vaccines encoding 4-1BBL in conjunction with TAAs [9,10,11]. These types of vaccines are cheap to produce, easy to construct, and have a long shelf life, plus they can be quickly manufactured for preclinical trials. However, there are some limitations, such as a local administration route that only transfects adjacent cells and a low transfection rate, resulting in the need for multiple immunizations to elicit an effective antitumor response [12,13]. The development of vaccines for HPV has been widely studied due to the clinical interest for population health and because it is a classic model of a virus-induced tumor that can be developed in preclinical models under laboratory conditions for the study of specific antigen responses. This has allowed the development and commercial distribution of Gardasil and Cervarix vaccines, which are virus-like particles (VLP) used as prophylactic vaccines in teenagers, principally in order to prevent infection with HPV 16 and 18 serotypes and to reduce the odds of developing HPV-associated tumors [14]. Nevertheless, it is still necessary to develop therapeutic vaccines to counter active tumor growth and to propose new strategies for TAA vaccine designs able to elicit specific antitumor responses.

Adenoviruses are one of the most efficient in vivo gene delivery systems due to their infective capacity, allowing transfection and gene expression on adjacent and distant cells after local administration [15]. Replication-defective adenovirus vectors hold gene deletions essential for viral replication (E1a and E1b), allowing them to be safe for clinical use, and these modifications are generated through homologous recombination in cells that express the E1 gene, such as HEK-293. They were initially used for their immunogenic properties for delivering genes directly to tumor cells, inducing an antitumor immune response [16,17]. Oncolytic adenoviruses possess the ability to selectively replicate in tumor cells, inducing cell lysis. Furthermore, tumor cell debris can be phagocyted and processed by APCs, which can potentiate the antigen-specific immune response that normally is obtained with conventional adenoviruses [18]. Therefore, oncolytic adenoviruses are particularly attractive for antitumor gene therapy.

Herein, in this study, we report the design and characterization of an oncolytic adenovirus expressing the SP/SA/E7/4-1BBL fusion gene cloned in an oncolytic adenovirus system to generate more efficient and selectively antitumor gene therapy.

## 2. Materials and Methods

### 2.1. Adenoviruses

The oncolytic adenovirus (OAd) used in this study was synthesized by O.D.260 Inc. (Boise, ID, USA). This OAd has a 24-bp deletion in the E1A conserved region 2 (CR2), a 1222-bp-long *Bgl*II-*Mfe*I deletion in the E3 region, in which the E3 ADP, RIDα, RIDβ, and 14.7K genes are preserved, a hybrid Ad5/3 fiber, and a CMV-(SP)-SA-E7-4-1BBL-SV40 pA expression cassette inserted between the fiber gene and the E4 region.

### 2.2. DNA Constructs

The SP-SA-E7-4-1BBL sequence, including codon optimization and restriction sites, was designed by our laboratory team, as described previously [9], and was synthesized by GenScript (Piscataway, NJ, USA).

### 2.3. Cell Lines

The human embryonic kidney cell line (HEK293) (# CRL-1573) used for adenovirus propagation was grown in Dulbecco’s modified Eagle’s medium (DMEM) (# 10-013-CV) and supplemented with 10% fetal bovine serum (FBS). TC-1 cancer cell line (# CRL-2785, discontinued) is derived from C57BL/6 murine lung epithelial cells that have been co-transformed with HPV16 E6/E7 and c-Ha-Ras-oncogenes. TC-1 cells were grown in RPMI-1640 media (# 10-040-CV) and supplemented with 5% FBS (# A31606, Thermo Scientific, Waltham, MA, USA) and 400 ug/mL G418 (A1720, Sigma Aldrich, St. Louis, MO, USA). Meanwhile, the 3T3 cell line derived from spontaneously immortalized murine embryonic fibroblasts (# CRL-1658) was grown in DMEM (# 10-013-CV) and supplemented with 10% newborn calf serum (NBCS) (#16010159, Thermo Scientific). All cell lines were purchased from the American Type Culture Collection (ATCC) (Manassas, VA, USA) and maintained at 37 °C in a 5% CO_2_ atmosphere.

### 2.4. Mice

C57BL/6 mice were acquired from Circulo ADN (Mexico City, Mexico). They were acclimatized for 5–7 days after their arrival and remained in our barrier animal facility at the School of Medicine, Autonomous University of Nuevo Leon, under cycles of 12-h light/12-h dark cycle, with *ad libitum* access to food and water in transparent cages with sawdust at a temperature of 25–28 °C. The evaluation was done every other day to determine their health status, as well as changing bedding, food, and water every third day. All animal procedures were performed following institutional guidelines and the principles outlined in the National Institutes of Health Guide for the Care and Use of Laboratory Animals (NIH Publications No. 8023, revised 1978). This study was analyzed and approved by the Ethics Committee of the School of Medicine, Autonomous University of Nuevo Leon (Monterrey, NL, Mexico) (protocol No. HT18-00002).

### 2.5. In Vivo Therapeutic Vaccinations

Groups of seven mice received 5 × 10^4^ TC-1 cells in 100 µL of phosphate-buffered saline (PBS) 1X in the right flank by subcutaneous injection. Two weeks after the tumor implant, the mice were randomly separated into five groups to receive the corresponding treatment. The three groups of mice that were immunized with the different OAds were injected intratumorally with 2.5 × 10^8^ UI in 20 µL of PBS 1X two times, one per week. The group that was treated with the DNA construct received 1 µg of DNA using the gene gun system on the shaved abdominal skin, with 1 immunization weekly for two weeks. The last group was treated intratumorally with 20 µL of PBS 1X as a negative control two times, one per week. Tumor progression was evaluated by measuring the tumor diameter three times per week using a digital caliper, and tumor volume was calculated by using the following formula: tumor volume = (tumor minor diameter^2^) × (tumor major diameter)/2. For survival analysis, all tumor-bearing mice were euthanized when tumors reached 1800 mm^3^ in tumor volume or earlier if ulceration was present or mice showed signs of discomfort.

### 2.6. Immunofluorescence

Cells (5 × 10^4^) were seeded overnight over 8 mm coverslips in a 24-well plate. The next day, the cells were infected with a multiplicity of infection (MOI) of 40 (for HEK-293) or 400 (for TC-1 and 3T3). After 16 h post-infection, cells were washed with cold PBS 1X, fixed with cold methanol for 10 min at −20 °C, and permeabilized with cold acetone for 30 s. Cells were washed with PBS 1x, blocked with 3% horse serum for 1 h at 4 °C, and incubated with a mix of anti-E7 monoclonal antibody (NM2) (Cat# sc-65711, Santa Cruz Biotechnology, Dallas, TX, USA) and anti-calnexin (H-70) (Cat# sc-11397, Santa Cruz Biotechnology) at a 1:500 dilution for 12 h at 4 °C. Next, cells were washed again and incubated for 2 h with goat anti-mouse IgG (H + L) CF594 and goat anti-rabbit CF488A IgG (H + L) (Santa Cruz Biotechnology). Coverslips were washed and mounted with Vectashield antifade mounting medium with 4′,6-diamidino-2-phenylindole (DAPI) (Vector Laboratories, Burlingame, CA, USA).

### 2.7. Western Blot Analysis

HEK-293 cells (5 × 10^5^) were seeded overnight in a 6-well plate. Next, they were infected with different MOIs from 5 to 40, incubated for 48 h, and processed as follows. Cells were harvested for the radioimmunoprecipitation assay buffer (RIPA buffer) lysis protocol. Cell lysates were quantified with the Pierce BCA protein kit (Thermo Scientific). A total of 25 µg of total proteins were electrophoresed on 10% SDS-polyacrylamide gels and transferred to PVDF membranes (GE Healthcare Life Sciences, Pittsburgh, PA, USA). Membranes were blocked with 10% skim-milk and then incubated with mouse anti-E7 monoclonal antibody (NM2) (sc-65711, Santa Cruz Biotechnology) and mouse anti-β-actin monoclonal antibody (A2228, Sigma-Aldrich). All washing steps were performed using TBS-Tween 1x. Next, membranes were incubated with secondary antibody anti-mouse HRP (1:5000, Bio-Rad Laboratories Inc., 170-6516) and developed with Supersignal West Pico Chemiluminescent Substrate kit (Thermo Fisher Scientific Inc., Waltham, MA, USA). For reproved incubations, membranes were stripped with stripping buffer 1x pH 2.2 and incubated in 10% skim-milk.

### 2.8. Viral Titration

To estimate the viral particle number present in the crude extracts, we performed the MOI calculation protocol as reported in [19]. In a 6-well plate with 1 × 10⁶ HEK-293 cells/well, the medium was completely removed and 500 µL of fresh medium was added, followed by the addition of 25, 50, 100, 150, and 200 µL of the crude extract to be tested. Cells were incubated for 3 h at 37 °C, 5% CO₂; subsequently, 1.5 mL of fresh medium was added to each well and incubated for 72 h. After that, cells were visually analyzed for cytopathic effect (CPE), where the minimum extract crude volume that produced CPE was considered as an MOI of 20. Uninfected cells were used as a negative control for CPE.

### 2.9. MTT and Violet Crystal Assays for Cell Viability

For MTT assays, 5 × 10^3^ cells/well were seeded in a 96-well plate in 200 µL of medium and incubated overnight for cell adherence. Next, 100 µL of medium/well was replaced with 200 µL of fresh media containing the indicated MOI and incubated for 72 h at 37 °C. Then, 30 µL of MTT reagent (5 mg/mL) was added to each well and incubated until precipitate formation was observed (~2 h). Finally, the medium was removed, 100 µL of DMSO was added for crystal solubilization, and the absorbance was read at 595 nm on the iMark Plate Reader (Bio-Rad).

For the violet crystal assay, 5 × 10⁴ cells/well were seeded in a 24-well plate and incubated overnight for cell adherence. Next, the medium was replenished with 2 mL of fresh media containing the indicated MOI and incubated for 72 h at 37 °C. The medium was removed, and 200 µL of 1% crystal violet-methanol was added and incubated for 20 min at RT. Then, the dye was removed and rinsed 4–5 times with ddH2O, and the plate was allowed to dry completely. Subsequently, 400 µL of methanol was added and left to incubate for 20 min at RT, stirring occasionally. Finally, the absorbance was read at 595 nm on the iMark Plate Reader (Bio-Rad). Uninfected cells were used as the negative control and considered as 100% cell viability.

### 2.10. Statistical Analysis

One-way ANOVA, followed by Dunnett’s multiple comparisons test, was performed to determine differences in cell viability across different treatments in the violet crystal assay. Two-way ANOVA, followed by Tukey’s multiple comparisons test, was performed to determine differences in cell viability across different treatments in the MTT assay. Both were performed using GraphPad Prism software version 6 (GraphPad Software, La Jolla, CA, USA). Differences with *p* ≤ 0.05 were considered significant (ns *p* > 0.05, * *p* ≤ 0.05, ** *p* ≤ 0.01, *** *p* ≤ 0.001, **** *p* ≤ 0.0001). All assays were performed at least twice, each one with triplicate data.

## 3. Results

### 3.1. SP/SA/E7/4-1BBL Protein Expression through Oncolytic Adenovirus Infection

In order to demonstrate whether the oncolytic adenovirus (OAd) expresses the SP/SA/E7/4-1BBL recombinant protein, HEK-293 cells were infected with the OAd at different multiplicity of infection (MOI) concentrations. After 72 h, cell lysates were analyzed by Western blot. Western blot analysis revealed a 55 kDa signal when using a monoclonal antibody against the E7 antigen, which corresponds to the expected molecular weight of the SP/SA/E7/4-1BBL recombinant protein (Figure 1a). Moreover, a dose-dependent increase in signal intensity was visualized with higher MOI concentrations, and in order to estimate the fold increase between each MOI concentration, a densitometric analysis was performed using the coefficient between SP/SA/E7/4-1BBL and actin (Figure 1b). The results show that at an MOI concentration of 5–20, there was no significant difference; therefore, it was used at an MOI of 40, which represents a 2-fold increase in the expression of the recombinant protein compared to actin. (Appendix A)

#### 3.1.1. Subcellular Localization of SP/SA/E7/4-1BBL Recombinant Protein

Since the SP/SA/E7/4-1BBL protein contains the signal peptide (SP) from human calreticulin (CRT), the recombinant protein is expected to be targeted to the endoplasmic reticulum (ER). To demonstrate this, an immunofluorescence assay was performed on infected HEK-293 (Figure 2a), TC-1 (Figure 2b), and NIH/3T3 (Figure 2c) cells. Once the cytopathic effect was observed, immunofluorescence was performed using antibodies against E7 and calnexin. The results reveal that the E7 signal (red) shows a perinuclear pattern that overlaps with the calnexin signal (green). In conclusion, we observed co-localization of both signals, demonstrating that our protein of interest is located in the ER.

#### 3.1.2. The Recombinant Oncolytic Adenovirus Displays Antitumor Activity In Vitro 

To prove that oncolytic adenovirus (OAd) expressing SP/SA/E7/4-1BBL can lyse tumor cells, we proposed infecting the tumor TC-1 cell line at different MOI concentrations for 72 h. It has been reported that some mouse cells are semi-permissive to OAd infection; therefore, we opted to increase the MOI concentration to 10- to 100-fold [20,21]. Cell viability was observed as detachment and decreased cell density, and the cytopathic effect was evaluated with the crystal violet assay (Figure 3). The results show that cell viability decreased in a dose-dependent manner: 97% at an MOI of 500, 94% at an MOI of 1000, 44% at an MOI of 2500, and 30% at an MOI of 5000. Moreover, the last two MOI concentrations were statistically significant (*p* < 0.001), thus confirming its ability to lyse tumor cells.

#### 3.1.3. The Cell Killing Effect of OAd Is Specific to Tumor Cells

Next, in order to demonstrate that the oncolytic effect of OAd expressing SP/SA/E7/4-1BBL is specific to tumor cells, TC-1 and NIH/3T3 cells were infected with OAd at different MOI concentrations for 72 h. Light microscopy revealed detachment and decreased cell density in infected TC-1 cells (cell count 58% at MOI 250, 36% at MOI 2500, and 29% at MOI 5000 with respect to cells without infection) (Figure 4a). This was in contrast to the non-tumor cell line NIH/3T3, which was unaffected by OAd (cell count 105% at MOI 250, 104% at MOI 2500, and 133% at MOI 5000 with respect to cells without infection), corroborated by quantification using Image J Cell Counter (Figure 4b). To quantify this effect, an MTT assay was performed to evaluate cell viability. At 72 h post-infection, TC-1 cells infected with OAd at MOI concentrations of 2500 and 5000 displayed a 43% (*p* < 0.05) and 67% (*p* < 0.01) decrease in cell viability, respectively, when compared to OAd-infected NIH/3T3 cells at the same MOI concentrations (Figure 5).

#### 3.1.4. The Recombinant Oncolytic Adenovirus Exhibits Antitumor Efficacy In Vivo

Next, to corroborate the oncolytic in vitro effect previously observed, we performed a therapeutic antitumor assay in a TC-1 mouse cancer model. Six- to 8-week-old C57BL/6 female mice were challenged with 5 × 10^4^ TC-1 cells injected in the right flank by subcutaneous injection. At 14 days after the tumor challenge (average tumor volume at 198.1 mm^3^ ± SEM 26.4), mice were intratumorally injected with OAd expressing SP-SA-E7-4-1BBL, SA-E7-4-1BBL, or SP-SA-4-1BBL at a concentration of 2.5 × 10^8^ UI. As a positive reference control, one group of mice was administered 1 µg of DNA constructs on shaved abdominal skin through the gene gun system, while another group of mice was injected with PBS 1X as a negative control. All groups received a weekly dose for two weeks. Tumor growth was monitored three times per week. By day 10 after the first treatment dose, we found tumor growth suppression in mice that were immunized with SP-SA-E7-4-1BBL (OAd) (*p* ≤ 0.0001) and SP-SA-E7-4-1BBL (DNA construct used as a positive reference control) (*p* ≤ 0.001), compared with the negative control (Figure 6a). Moreover, at the same time, the negative control and SP-SA-4-1BBL (OAd) mice groups reached tumor volume endpoint criteria, while the rest of the groups maintained 100% survival (Figure 6b). From this point to the end of the study (week 4), there was no statistically significant difference observed between the three remaining groups (*p =* 0.4910). Overall, these results demonstrate that the oncolytic adenovirus expressing SP-SA-E7-4-1BBL and SA-E7-4-1BBL is as effective in delaying tumor progression as the DNA vaccine expressing SP-SA-E7-4-1BBL.

## 4. Discussion

To guarantee an efficient infection with oncolytic adenovirus, it is important to consider the mechanism of cellular internalization of the virus and the species of origin of the cell lines used in the assays. The coxsackie-adenovirus (CAR) receptor is the primary receptor used by the OAd to attach to the cell surface, followed by an interaction with cellular integrins. Therefore, in cells that have low or no CAR expression, the internalization pathway depends exclusively on the integrins [22]. Human cells express CAR, so they can be infected with a low MOI. On the contrary, murine cells have a low expression of this receptor, and therefore, elevated MOI concentrations should be used to assure their transduction. In the present study, TC-1 and NIH/3T3 murine cell lines had to be infected with an MOI of up to 5000 to achieve a cytopathic effect.

All adenovirus-based vectors are derived from the human adenovirus serotype 5, which makes these viruses unable to produce progeny in murine cells [23,24]. Nevertheless, human adenoviruses in murine cells can produce the necessary viral proteins by regulating the transcription-translation machinery of infected cells, even though OAds are not able to produce their progeny efficiently [25]. This was demonstrated by the ability of OAds to direct the expression of the SP/SA/E7/4-1BBL protein in tumor and normal murine cells.

Since the SP/SA/E7/4-1BBL recombinant protein contains a signal peptide (SP) from human calreticulin (CRT), we demonstrated its localization in the ER lumen. Previously, we have demonstrated that adding a signal peptide (SP) and ER-retaining signal (KDEL) to the E7 antigen confers a more potent antitumor effect [26,27]. However, in this case, our interest was for this protein to be sent only to the secretory pathway without being held in the ER. Therefore, only the SP was added, and this would also help for the correct protein folding for 4-1BBL.

Recently, we proved that the SP-SA-E7-4-1BBL DNA vaccine exhibited outstanding therapeutic and prophylactic effects against HPV-16 E7-expressing TC-1 tumors [9]. DNA vaccines offer several advantages, such as easy use at low cost and especially their ability to stimulate cellular and humoral responses. Nevertheless, this effect has been limited to preclinical models [28]. Replication-competent adenoviruses have been broadly used in cancer gene therapy, and they are designed to replicate preferentially in tumor cells and destroy them through the natural lytic process of viral replication [29]. Therefore, we decided to use a combination of these two mechanisms: the expression of a transgene that has previously demonstrated an antitumor effect and the specific lysis of tumor cells by oncolytic adenoviruses.

In this study, we show that oncolytic adenovirus has a cell killing effect on TC-1 tumor cells. It has been reported that even in the absence of viral progeny production, excessive production of viral proteins can result in the death of infected tumor murine cells, primarily by the activation of autophagy, although it is capable of activating other types of cell death [30,31,32].

After we showed that OAd is capable of infecting NIH/3T3 non-tumor cells and expressing the SP/SA/E7/4-1BBL protein, we evaluated whether cell viability was affected. Therefore, we performed a viability assay on both murine cell lines. The MTT assay results revealed that the non-tumor cell line had no decrease in its viability, and instead, that there was an increase in metabolic viability in infected cells compared to the infection-free control.

To survive and replicate in cells, viruses must take control of the various cellular organelles involved in defense and immune processes. Once inside the host cell, they modulate cell signalization pathways and organelles, including mitochondria, and use them for their own survival [33]. As has been reported, increased metabolic viability may be due to hyperactive mitochondria or an increase in mitochondrial mass, since the conversion of MTT into formazan occurs mainly in the mitochondria [34]. Therefore, a non-tumor cell line can cause an increment in mitochondrial activity during Ad infection, resulting in a false cell proliferation signal; otherwise, a tumor line would be affected and eventually lysed by OAd, causing a reduction of formazan formation.

All together, these results show that the oncolytic adenovirus used in this study can express the protein SP/SA/E7/4-1BBL, which is targeted to the ER lumen. Although this expression occurs in both tumor and non-tumor cells, the cell killing effect is restricted to tumor cells. To corroborate this specific antitumor effect in vivo, we demonstrate in this study that the administration of OAd encoding 4-1BBL fused to E7 antigen in mice with established tumors resulted in the suppression of tumor growth, as well as in 100% survival regarding the reference control. When we compare such effects with the administration of the gene construct as a DNA vaccine, we observed that the antitumor effect is equivalent. Therefore, it was possible to improve delivery and specificity without compromising the previously demonstrated antitumor effect [9]. Further studies should analyze the safety and biodistribution of recombinant adenovirus, as well as correlate the mechanisms involved in the antitumor effect. It should be noted that although our study focused on the E7 antigen and an HPV-induced mouse cancer model, this strategy can be translated to a variety of tumor-associated antigens for cancer gene therapy.

## 5. Conclusions

In the present study, we report the design of an oncolytic adenovirus expressing the SP/SA/E7/4-1BBL fusion gene, which is capable of infecting murine cancer and normal cells, and the expressed protein was able to be targeted to the endoplasmic reticulum. Most importantly, OAd induced a cell killing effect that is specific to cancer cells. Moreover, OAd treatment induced a potent antitumor effect in a mouse TC-1 cancer model.

## Figures and Tables

**Figure 1 vaccines-09-00149-f001:**
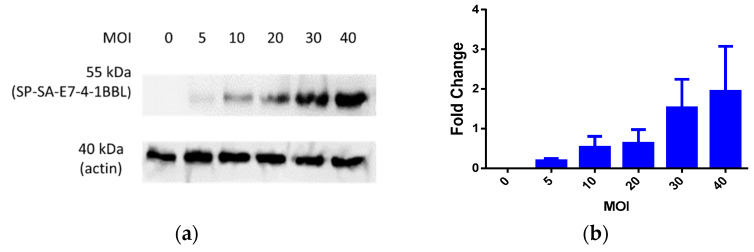
Detection of SP/SA/E7/4-1BBL protein expression in HEK-293 cells. (**a**) Fifty-five kDa bands corresponding to the protein of interest at MOI concentrations of 0, 5, 10, 20, 30, and 40; normalized results with actin as an endogenous control. (**b**) Densitometry graph where greater expression of SP/SA/E7/4-1BBL protein is observed in a dose-dependent manner, relative to the expression of actin. MOI, multiplicity of infection.

**Figure 2 vaccines-09-00149-f002:**
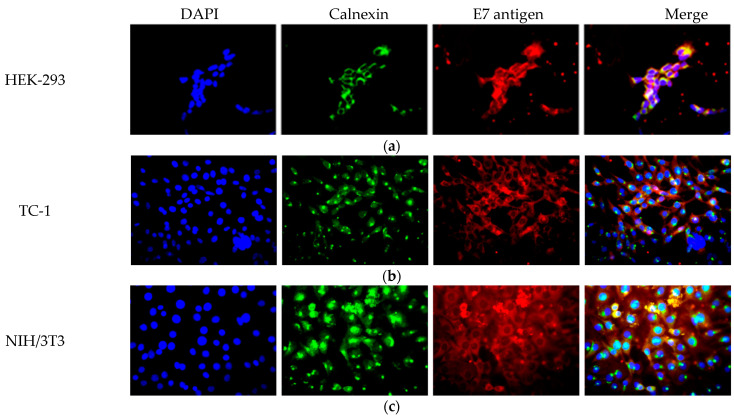
Subcellular localization of SP/SA/E7/4-1BBL recombinant protein in (**a**) HEK-293, (**b**) TC-1, and (**c**) NIH/3T3 cell lines. Blue channel DAPI signal for the nucleus, red channel signal for E7 antigen, green channel signal for calnexin; merge (yellow stain) shows co-localization of signals for E7 and calnexin. DAPI, 4′,6-diamidino-2-phenylindole.

**Figure 3 vaccines-09-00149-f003:**
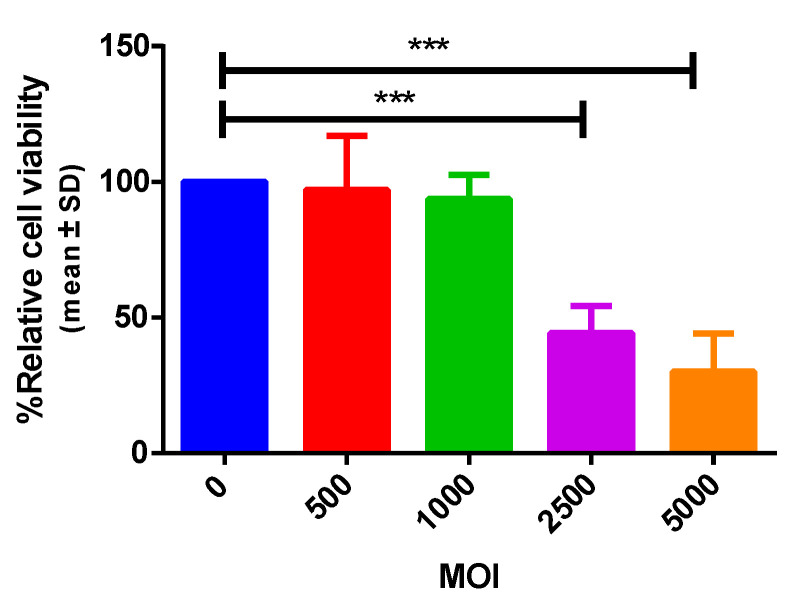
In vitro antitumor effect of oncolytic adenovirus expressing SP/SA/E7/4-1BBL. Cell viability percentage of TC-1 cells was measured by crystal violet assay after infection with adenoviruses at different MOIs, which decreases while MOI increases, compared to infection-free control. *** (*p* < 0.001).

**Figure 4 vaccines-09-00149-f004:**
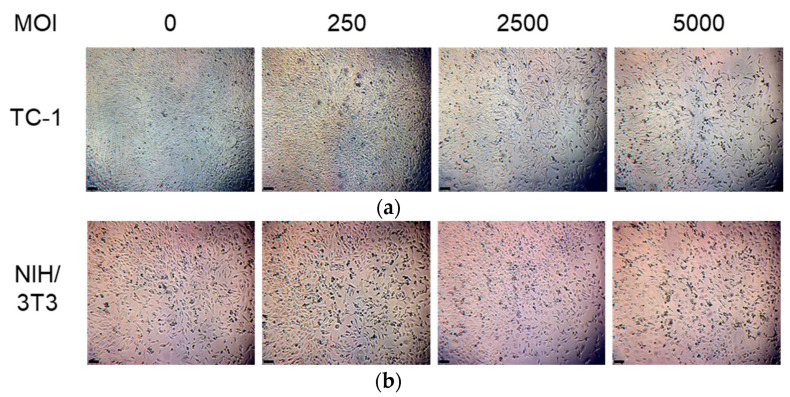
Specific oncolytic effect against tumoral cells. Evaluation of cellular density by light-field microscopy of TC-1 (**a**) and NIH/3T3 (**b**) cells after infection with oncolytic adenoviruses. Both cell lines were infected with different MOIs and incubated for 72 h. The TC-1 tumor cell line displays detachment and a decrease in cell density as MOI increases; nevertheless, the NIH/3T3 non-tumor cell line was unaltered. 1 = 0 MOI, 2 = 250 MOI, 3 = 2500 MOI, 4 = 5000 MOI. Scale bars 100 µm.

**Figure 5 vaccines-09-00149-f005:**
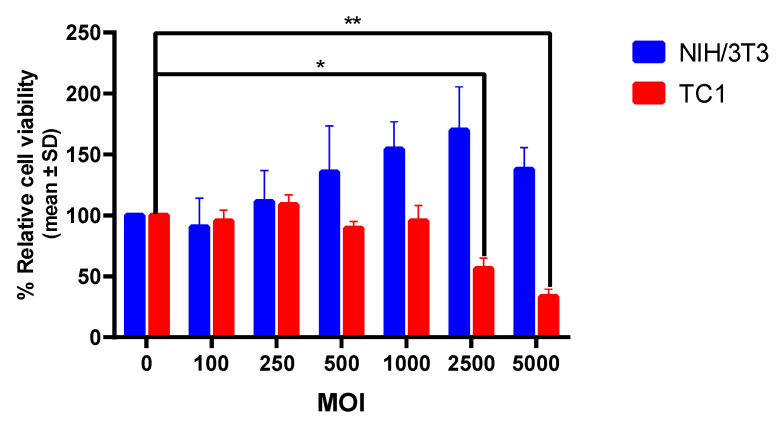
Cell viability percentage graph after 72 h post-infection with recombinant oncolytic adenoviruses. The TC-1 cell line displayed 43% mortality at an MOI of 2500 * (*p* < 0.05) and 67% at an MOI of 5000 ** (*p* < 0.01), while the non-tumor NIH/3T3 cell line exhibited no significant changes.

**Figure 6 vaccines-09-00149-f006:**
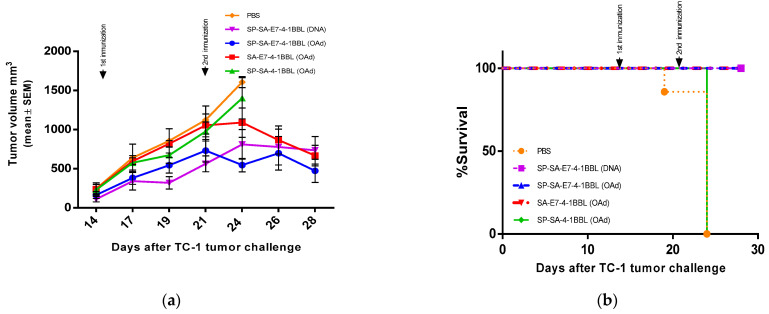
In vivo therapeutic antitumor effect of oncolytic adenovirus. (**a**) Tumor volume was measured after the first administration of oncolytic adenovirus DNA or PBS at 14 days after tumor challenge with 5 × 10^4^ TC-1 cells in the right flank by subcutaneous injection. Groups (*n* = 7) of 6- to 8-week-old C57BL/6 mice were injected intratumorally with 2.5 × 10^8^ UI of oncolytic adenovirus two times (one per week). The DNA construct was administrated on shaved abdominal skin with 1 µg of DNA using a gene gun system. PBS was used as a negative control. Tumor volumes are represented by mean ± SEM. Mice were euthanized when the tumor volume was higher than 1800 mm^3^. (**b**) Survival graph after TC-1 tumor challenge. Two-way ANOVA and post-hoc Tukey’s tests were performed.

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
