# Peer review of "An Oncolytic Adenovirus Encoding SA-4-1BBL Adjuvant Fused to HPV-16 E7 Antigen Produces a Specific Antitumor Effect in a Cancer Mouse Model"

_vaccines, 2021, doi:10.3390/vaccines9020149_

Round 1

Reviewer 1 Report

The study is based on new developments. The results provided here indicate that in the present study were report the design of an oncolytic adenovirus expressing the SP/SA/E7/4-1BBL gene fusion which is capable of infecting murine cancer and normal cells, and the expressed protein was able to be targeted to the Endoplasmic Reticulum.

It was determined that OAd induced a cell killing effect that is specific to cancer cells. OAd treatment induced a potent antitumor effect in a mouse TC-1 cancer model.

This article is very interesting and relevant for development of new vaccines for cancer.

The conclusions are consistent with the evidence and arguments presented.

The authors address the main question posed.

Author Response

Moderate English changes required. 

We will request MDPI English Editing Services if the manuscript is accepted for publication.

Reviewer 2 Report

In this report, Alejandra et. al., constructed an oncolytic adenovirus with modulatory molecule SA-4-1BBL on E7 showed the anti-tumor effect in vitro and in vivo experiments. 

There is insufficient demonstration of the specifically designed for oncolytic adenovirus on both tumor cells and immune cells. The following concerns should be addressed before further consideration of this manuscript.

Major:

  • WT control should be included to identify the effect of SA/SP/4-1BBL design, otherwise, all this effect is based only on oncolytic adenovirus alone.
  • 4-1BBL could provide a strong activation signaling for immune cells, author should design experiments to show the 4-1BBL effect on immune cells, such as co-culture with T or NK cells. 
  • The experiments of ER location facilitate oncolytic adenovirus’ anti-tumor effect could further strengthen the paper.

Minor points:

Fig 2 Virus in TC-1 looks like not only on ER like HEK-293 but spread over the cytoplasm. Please provide quantification of this effect.

Fig 4 Quantification of this effect, the microscopy could not tell the significant difference.

Fig 5 Cell viability or proliferation assay? Since the viability for cells could reach nearly 200%

Author Response

Major points

Point 1: WT control should be included to identify the effect of SA/SP/4-1BBL design, otherwise, all this effect is based only on oncolytic adenovirus alone.

Response 1: In our previous work on DNA vaccines [1] it was shown that the use of 4-1BBL alone does not induce an antitumor effect in our murine cancer model, similar to that observed when using the empty vector and E7wt construct, on prophylactic and therapeutic assay; so it was decided to choose to use SA/SP/4-1BBL as a base control to reduce the murine groups.

On this paper we used an oncolytic adenovirus capable of lysing tumor cells as demonstrated at in vitro assays. Although, it was expected that oncolytic activity could decrease tumor growth, in vivo assay showed that SA/SP/4-1BBL cannot decrease tumor growth, such as occurred in PBS control group. This could be that in our model we used low concentrations of adenovirus that were surpassed by tumor cell cycle. Nevertheless, when we incorporate E7 antigen into OAd constructs (SP/SA/E7/4-1BBL and SA/E7/4-1BBL), it was possible to detect a reduction in the speed of tumor growth after the second immunization and subsequently a non-significance higher tumor reduction than our reference control: SP/SA/E7/ 4-1BBL DNA vaccine. Therefore, in the experimental conditions of our murine cancer model, we state that observed anti-tumor effect is due co-expression of E7 antigen and expression 4-1BBL that triggers an anti-tumor response rather than an oncolytic based effect. (n=7).

Point 2: 4-1BBL could provide a strong activation signaling for immune cells, author should design experiments to show the 4-1BBL effect on immune cells, such as co-culture with T or NK cells. 

Response 2: This work has not included 4-1BBL effect tests, however, if gaining grants, it is proposing to develop work focused on the study of 4-1BBL-induced signaling and other ligands in vitro in T cells culture. 

Point 3: The experiments of ER location facilitate oncolytic adenovirus’ anti-tumor effect could further strengthen the paper.

Response 3: Our group study has reported that antigens delivered to ER induce a higher specific antigen response, such as IFN-gamma release [1–4]This effect has been related to ER-stress responses that trigger mechanisms that could enhance antigen processing and presentation to immune cells.

In fact, as reviewer pointed, and our in vivo results shows that when an SP signal is added (SP-SA-

E7-4-1BBL) provides an early and sustained antitumor effect compared when SP is absent (SA-E7-4-1BBL). Therefore, ER could participate in anti-tumor effect and for our group is a future goal to describe ER-related mechanisms are triggered when OAd infection occurs.

Minor points:

Fig 2 Virus in TC-1 looks like not only on ER like HEK-293 but spread over the cytoplasm. Please provide quantification of this effect.

Response Fig 2: The reviewer comments to quantify this effect, we understand that refers to the number of events of E7-Calnexin signal overlapping to determine if it is inside ER. In previous research [4,5] a confocal microscope was available and was used to analyze the effectiveness of the SP signal, the same sequence used in the present paper for ER delivering. As a result, it was got that the fusion protein used (E6E7) was effectively sent to the ER by quantifying the average fluorescence recorded overlapping the signal for calnexin (ER).

In the present work, epifluorescence microscopy is used, so we cannot adjust to a cut plane and specifically analyze the signal at a given site. In addition, since the TC-1 line is a tumor and that we showed that it was lysed by OAd, it is hypothesized that due ER-targeting, onset of cellular stress and lytic effect could be perceived as a spread signal under an epifluorescence microscope. However, we detected overlapping signal persistent in all infected cells. Also, it was ruled out that the pattern observed could be due to non-specificity binding of antibodies.

Fig 4 Quantification of this effect, the microscopy could not tell the significant difference.

Response Fig 4: The light microscopy image only had the purpose of qualitatively demonstrating the decrease in cellularity and the morphological changes of cell culture at the different MOIs. However, the reviewer's observation was noted, and an ImageJ quantification (using ImageJ Cell Counter) was performed on the images presented, as well as other images from experiments developed with the same variables. The results were added to the article text as a previous to the feasibility analysis by MTT (line 235-239).

Fig 5 Cell viability or proliferation assay? Since the viability for cells could reach nearly 200%

Response Fig 5: It is a cell viability assay. The MTT technique quantifies the redox function of the mitochondrial by transformation of MTT to formazan (colored compound), which indirectly allows to infer cell viability status. This technique has been used as a standard cell culture viability test [6]. To spread, viruses require the use of cell vesicles and cause an overstimulation of the mitochondria redox activity [7]. Therefore, a non-tumor, can course an increment in mitochondrial activity during Ad infection, resulting in a false cell proliferation signal; otherwise, a tumor line would be affected and eventually lysed by OAd causing a reduction of formazan formation. The above was mentioned in the discussion (line 370-375), however pertinent adjustments were made to improve text writing (line 373-375).

  1. Garza-Morales, R.; Perez-Trujillo, J.J.; Martinez-Jaramillo, E.; Saucedo-Cardenas, O.; Loera-Arias, M.J.; Garcia-Garcia, A.; Rodriguez-Rocha, H.; Yolcu, E.; Shirwan, H.; Gomez-Gutierrez, J.G.; et al. A DNA Vaccine Encoding SA-4-1BBL Fused to HPV-16 E7 Antigen Has Prophylactic and Therapeutic Efficacy in a Cervical Cancer Mouse Model. Cancers 2019, 11, doi:10.3390/cancers11010096.
  2. Martínez-Puente, D.H.; Pérez-Trujillo, J.J.; Gutiérrez-Puente, Y.; Rodríguez-Rocha, H.; García-García, A.; Saucedo-Cárdenas, O.; Montes-de-Oca-Luna, R.; Loera-Arias, M.J. Targeting HPV-16 Antigens to the Endoplasmic Reticulum Induces an Endoplasmic Reticulum Stress Response. Cell Stress Chaperones 2019, 24, 149–158, doi:10.1007/s12192-018-0952-8.
  3. Loera-Arias, M.J.; Martínez-Pérez, A.G.; Barrera-Hernández, A.; Ibarra-Obregón, E.R.; González-Saldívar, G.; Martínez-Ortega, J.I.; Rosas-Taraco, A.; Villanueva-Olivo, A.; Esparza-González, S.C.; Villatoro-Hernandez, J.; et al. Targeting and Retention of HPV16 E7 to the Endoplasmic Reticulum Enhances Immune Tumour Protection. J. Cell. Mol. Med. 2010, 14, 890–894, doi:10.1111/j.1582-4934.2009.00934.x.
  4. Perez-Trujillo, J.J.; Garza-Morales, R.; Barron-Cantu, J.A.; Figueroa-Parra, G.; Garcia-Garcia, A.; Rodriguez-Rocha, H.; Garcia-Juarez, J.; Muñoz-Maldonado, G.E.; Saucedo-Cardenas, O.; Montes-De-Oca-Luna, R.; et al. DNA Vaccine Encoding Human Papillomavirus Antigens Flanked by a Signal Peptide and a KDEL Sequence Induces a Potent Therapeutic Antitumor Effect. Oncol. Lett. 2017, 13, 1569–1574, doi:10.3892/ol.2017.5635.
  5. Pérez-Trujillo, J.J.; Robles-Rodríguez, O.A.; Garza-Morales, R.; García-García, A.; Rodríguez-Rocha, H.; Villanueva-Olivo, A.; Segoviano-Ramírez, J.C.; Esparza-González, S.C.; Saucedo-Cárdenas, O.; Montes-de-Oca-Luna, R.; et al. Antitumor Response by Endoplasmic Reticulum-Targeting DNA Vaccine Is Improved by Adding a KDEL Retention Signal. Nucleic Acid Ther. 2018, 28, 252–261, doi:10.1089/nat.2017.0717.
  6. Rai, Y.; Pathak, R.; Kumari, N.; Sah, D.K.; Pandey, S.; Kalra, N.; Soni, R.; Dwarakanath, B.S.; Bhatt, A.N. Mitochondrial Biogenesis and Metabolic Hyperactivation Limits the Application of MTT Assay in the Estimation of Radiation Induced Growth Inhibition. Sci. Rep. 2018, 8, doi:10.1038/s41598-018-19930-w.
  7. Anand, S.K.; Tikoo, S.K. Viruses as Modulators of Mitochondrial Functions. Adv. Virol. 2013, 2013, doi:10.1155/2013/738794.

Reviewer 3 Report

In the present paper, the authors assessed the possible use of an oncolytic adenovirus expressing the SP/SA/E7/4-1BBL gene fusion cloned on an oncolytic adenovirus system to provide an efficient and selective antitumor gene therapy. The results showed that the model only works at extremely high MOI values (2500, 5000). The authors should clarify some important points before consideration:

1) Is there any possibility of adverse effects at such high MOI concentrations?

2) A dose-dependent study on mouse model may be useful.

Author Response

Point 1: Is there any possibility of adverse effects at such high MOI concentrations?

Response 1: The doses administered to the murine model were selected after a review of previous preclinical studies that involve this type of delivery system and constitute a safe dose [1–3]. The administered dose corresponds to an MOI of 5000 concerning the number of implanted tumor cells (5 ˟ 104). In the in vitro tests, no cellular toxicity was observed at this dose; however, the general condition of the mice was monitored during treatment. Parameters such as weight, appetite, and physical activity did not show any alterations.

Point 2: A dose-dependent study on mouse model may be useful.

Response 2: In this model, we decided to translocate the in vitro model to the in vivo model to guarantee the observed antitumor response. Considering the ability of OAd to replicate and lyse tumor cells, new virions would be released and amplifying the effect so dose-dependent study would be considered in future. Nevertheless, our actual objective was to use sub-optimal doses rather than tumor-eradicating doses, this allowed us to track tumor growth and infers OAd in vivo activity when 4-1BBL is used with tumor-associated antigens (TAAs) such as E7 antigen.

  1. Atherton, M.J.; Stephenson, K.B.; Nikota, J.K.; Hu, Q.N.; Nguyen, A.; Wan, Y.; Lichty, B.D. Preclinical Development of Peptide Vaccination Combined with Oncolytic MG1-E6E7 for HPV-Associated Cancer. Vaccine 2018, 36, 2181–2192, doi:10.1016/j.vaccine.2018.02.070.
  2. Dai, B.; Roife, D.; Kang, Y.; Gumin, J.; Rios Perez, M.V.; Li, X.; Pratt, M.; Brekken, R.A.; Fueyo-Margareto, J.; Lang, F.F.; et al. Preclinical Evaluation of Sequential Combination of Oncolytic Adenovirus Delta-24-RGD and Phosphatidylserine-Targeting Antibody in Pancreatic Ductal Adenocarcinoma. Mol. Cancer Ther. 2017, 16, 662–670, doi:10.1158/1535-7163.MCT-16-0526.
  3. Zhang, L.; Hedjran, F.; Larson, C.; Perez, G.L.; Reid, T. A Novel Immunocompetent Murine Model for Replicating Oncolytic Adenoviral Therapy. Cancer Gene Ther. 2015, 22, 17–22, doi:10.1038/cgt.2014.64.

Round 2

Reviewer 2 Report

The authors have addressed my concerns adequately.

Reviewer 3 Report

The authors'response is satisfactory. The study can be published in its current form.